# Qualitative Research in Digital Era: Innovations, Methodologies and Collaborations

**Grzegorz Bryda** [1,*] and **António Pedro Costa** [2]

1   CAQDAS TM Lab, Institute of Sociology, Jagiellonian University, 31-007 Kraków, Poland
2   Research Centre on Didactics and Technology in the Education of Trainers, Department of Education and Psychology, University of Aveiro, 3810-193 Aveiro, Portugal; apcosta@ua.pt
*   Correspondence: grzegorz.bryda@uj.edu.pl

**Abstract:** The differentiation of contemporary approaches to qualitative data analysis can seem daunting even for experienced social science researchers. Especially when they move forward in the data analysis process from general analytical strategies used in qualitative research to more specific approaches for different types of qualitative data, including interviews, text, audio, images, videos, and so-called virtual data, by discovering the domain ontology of the qualitative research field, we see that there are more than twice as many different classes of data analysis methods as qualitative research methods. This article critically reflects on qualitative research and the qualitative computer data analysis process, emphasising its significance in harnessing digital opportunities and shaping collaborative work. Using our extensive analytical and research project experience, the last research results, and a literature review, we try to show the impact of new technologies and digital possibilities on our thinking. We also try to do the qualitative data analysis. The essence of this procedure is a dialectical interplay between the new world of digital technology and the classic methodology. The use of digital possibilities in qualitative research practices shapes the researcher's identity and their analytical and research workshop. Moreover, it teaches collaborative thinking and teamwork and fosters the development of new analytical, digital, and Information Technology (IT) skills. Imagining contemporary qualitative research and data analysis in the humanities and social sciences is difficult. Opening to modern technologies in computer-based qualitative data analysis shapes our interpretation frameworks and changes the optics and perception of research problems.

**Keywords:** CAQDAS; artificial intelligence; digital methods; digital skills; collaborative analysis

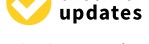



## 1. Introduction

In the modern era, technological advancements have dramatically transformed research and data analysis within the realm of social sciences. Big data, advanced statistical tools, digital platforms, and, notably, CAQDAS (Computer Assisted Qualitative Data Analysis Software) have broadened the scope of research possibilities. Tools such as NVivo, Atlas.ti, webQDA, and MAXQDA under the CAQDAS umbrella have revolutionised qualitative research. They enable scholars to efficiently organise, categorise and analyse vast amounts of textual, audio, and visual data. These software solutions offer nuanced coding mechanisms, facilitating a deeper understanding of the themes, patterns, and narratives present in the data. This systematic approach enhances the rigour and credibility of qualitative analyses and bridges the divide between qualitative and quantitative methods. Moreover, social media sentiment analysis and large-scale surveys provide researchers with more prosperous and diverse datasets. Machine learning and artificial intelligence techniques can now comb through these vast datasets precisely, identifying patterns and insights that were once beyond reach.

Additionally, data visualisation tools make complex data more understandable, promoting a more explicit interpretation and communication of the results. Navigating the

intricacies of the digital age, the intersection of technology and social sciences—as strengthened by tools like CAQDAS—promises to enhance our understanding of societies. This fusion ensures that we are equipped with more effective, data-driven solutions to address the multifaceted challenges faced by contemporary societies.

Emerging technologies are revolutionizing how researchers collaborate on projects, transcending social, cultural, and geographical boundaries. This new age of qualitative research underscores the unique challenges and opportunities, emphasizing the imperative for collaborative strategies. The CAQDAS leads these technological advancements. This essay delves into the opportunities and challenges CAQDAS presents, examining its pivotal role in collaborative research. Researchers must harness these tools since digital networks empower extensive expression and content dissemination. This essay contends that becoming proficient in qualitative research methodologies and computer-assisted data analysis is closely linked to grasping and applying new digital technologies and enhancing digital skills. As Costa and Costa (2017) suggested, conducting research in digital environments can offer fresh perspectives to traditional research methodologies.

Furthermore, we emphasise that technological advancements enable researchers to engage in real-time sharing and collaboration on the same project, regardless of geographical boundaries. Research inherently presents challenges and opportunities, making adopting collaborative strategies highly beneficial. This essay seeks to stimulate conversation around the benefits and challenges of collaborative research for researchers and how technology facilitates this collaborative approach. Indeed, the digital realm offers numerous avenues for expressing and disseminating information. It falls upon researchers to harness and navigate these abundant resources.

We aim to spotlight five pivotal areas in qualitative research that technological advancements have profoundly influenced. These areas represent the evolution of research methodologies and the promising future of digitally augmented research. At the forefront of this evolution is CAQDAS. This software provides researchers with a robust platform to analyse qualitative data meticulously and efficiently. As the backbone of digital qualitative research, CAQDAS symbolises the seamless marriage of technology with traditional research methodologies, ensuring precise and streamlined outcomes. Beyond tools, the digital era introduces new paradigms of managing information. "Datafication" pertains to converting various forms of information into a digital format. This process streamlines data analysis and ensures that qualitative researchers can easily handle extensive and complex datasets. The digital realm offers various innovative tools and methodologies that fundamentally alter the qualitative research landscape. From virtual interviews to AI-assisted transcription and beyond, the scope of what is possible in research has been expanded manifold, leading to more prosperous, more nuanced insights. One of the most profound shifts brought about by the digital age is the transformation of collaboration in research. Digital platforms have made real-time, global collaboration feasible. This interconnectedness fosters a more holistic and multidisciplinary approach to research, where varied perspectives converge to produce more prosperous outcomes. Finally, with the integration of cutting-edge technologies like virtual reality, augmented reality, and machine learning, the boundaries of what qualifies as "qualitative research" are being continually redefined. These innovations enable researchers to explore previously uncharted territories, producing more comprehensive and reflective findings. In conclusion, understanding these five areas is paramount as the lines between technology and qualitative research become increasingly intertwined. The future promises a closer synergy between digital tools and research methodologies, opening doors to deeper insights and revolutionary findings.

## 2. The CAQDAS R/Evolution: Bridging Technology with Tradition

Throughout the history of qualitative research, a paradigm-shifting r/evolution has been led by CAQDAS. This computer-aided analysis is a cornerstone of modern qualitative research, especially in content (textual) analysis and linguistic evaluations. The origins of data analysis software and the integration of computers in social sciences are rooted

in the latter half of the twentieth century (Brent and Anderson 1990; Fielding and Lee 1991). CAQDAS was primarily linked with leveraging technology for quantitative content analysis (Berelson 1984) and Grounded Theory (Glaser and Strauss 1965). Within this domain, debates over the use of computers in qualitative research are most heated and continue to evoke strong emotions. Yet, with the digital revolution of the past two decades, the surge of new information technologies, and the rise of digital media, CAQDAS has expanded its reach and has found broader applications. This expansion, fuelled by digital advancements, has paved the way for more general applications in traditional qualitative research, encompassing narrative methods, interviews, and textual and visual content analyses. A fundamental transition occurred when state-of-the-art technology began integrating with conventional research methodologies, marking a transformative phase in the ever-evolving qualitative research methodology landscape. Through this synthesis of technology and tradition, CAQDAS bridges traditional data analysis methods and the modern, digitised methods enabled by technological advancements between historical methodologies and the demands of contemporary practices. It harmoniously merges quantitative and qualitative research methods, enhancing the depth of data visualisation.

The social sciences, education, health and medicine, business, and the humanities are just a few of the fields that employ CAQDAS. For instance, qualitative researchers utilise it to gather, plan, and analyse substantial volumes of qualitative data from several teams in an international project (Robins and Eisen 2017) and support analytical awareness and reflexivity (Woods et al. 2016). Additionally, it is utilised to help with record-keeping, organise and manage qualitative data, give an audit trail, improve and show methodological rigour, and allow researchers to advance from theory creation to description (Bringer et al. 2006). CAQDAS plays a pivotal role in data management and the overarching research process within these disciplines. Its capabilities are vast, segmenting text, image, sound, and video into manageable units and semantically indexing them for in-depth analysis. The software suite includes functionalities ranging from coding, memoing, and paraphrasing to annotating, grouping, and network-building. Moreover, its tools aid in lexical searching, data retrieval, and comparisons, evolving into a comprehensive platform for researchers. Features like data visualisation, collaboration support across time and space, and facilitating quantitative and qualitative methodologies make CAQDAS a prominent tool in today's research. Notably, it assists in illustrating patterns and trends and aids qualitative researchers in visualizing code relations and creating semantic networks (Bryda 2019, 2020). These multifaceted utilities ensure a transparent research process, bolstering methodological rigour and empowering scholars in their analytical endeavours.

Contrary to the 1980s and 1990s, CAQDAS programs now include various functionalities dedicated to users in the humanities and social sciences. However, availability depends on the type of software. CAQDAS software is divided into three groups: licensed programs equipped with many advanced and exciting analytical functionalities, open-source tools that usually have a basic range of functionality, and online programs that are functionally advanced to varying degrees. These innovations have positioned CAQDAS at the forefront of the modern qualitative research revolution, striking a balance between the cherished methods of yesteryears and the technological potential of today. Their advantage, as opposed to licensed and open-source ones, is an extensive range of flexibility that enables the development or implementation of new algorithms and analytical techniques based on programming languages. Their advantage is the possibility of synchronous and asynchronous teamwork and combining qualitative and quantitative methods classes. The literature meta-analyses of the contemporary field of qualitative research (Qualitative Domain Ontology) show that the development of CAQDAS over the last three decades is reflected in the currently dominant narrative qualitative methodology (Bryda et al., forthcoming). The publications in the field of CAQDAS show that software development and its analytical functionalities tend towards the methods and procedures of textual and visual data content analysis. Conversely, an emerging trend integrates computer-aided qualitative data analysis with methods from the digital humanities, natural language processing, and text

mining (Bryda 2014; Bryda and Tomanek 2014). This convergence has given rise to the new interdisciplinary field of Digital Qualitative Sociology (Bryda 2014).

The rapid advancement of technology has reshaped many sectors, with qualitative research being prominently affected. Understanding and leveraging new digital technologies, alongside enhancing digital capabilities, are crucial for researchers in this domain. In qualitative research, it is essential to understand the latest digital technologies and skills. Tools like CAQDAS have significantly changed the way we approach qualitative research. Over the past years, the applications of CAQDAS have expanded, moving beyond its original purpose. However, it is critical to remember that traditional research methods remain vital even with these new digital tools. The two complement each other, enhancing the overall quality of research. Collaboratively cultivating and harnessing digital potential has become a hallmark of modern qualitative research methodology and data analysis. Digitality is pivotal for advancing the contemporary qualitative research methodology and CAQDAS. For researchers to excel in computer-assisted data analysis, proficiency in these fast-evolving digital tools and methods is imperative. The benefits of mastering digital technology are evident in the diverse applications and advantages of tools like CAQDAS. It is a platform that harmoniously integrates both old and new research methods. A robust understanding of digital skills is essential to use these tools to their fullest potential. Qualitative researchers must employ these digital tools and methodologies in the contemporary research landscape.

## 3. Digital Society, Methods, and Possibilities

Digitality is a foundational element in understanding the concept of a digital society. This term encapsulates modern societies' transformative shifts as they embrace and integrate information and communication technologies across daily life, including home, work, education, and leisure. As Lindgren (2022) notes, digital innovations are reshaping our societal structures, economic dynamics, and cultural landscapes at an unprecedented pace and magnitude. Digitality can be defined as the experience of living within a digital culture. This term is inspired by Nicholas Negroponte's book, "Being Digital," drawing parallels with the concepts of modernity and post-modernity (Negroponte 1996), especially with the Digital Society approach. The Digital Society, as defined by Schwarz (2021), is an emergent and interdisciplinary research domain that arises from integrating advanced technologies into our societal fabric and cultural norms. Central to this evolution is a need to grasp the profound shifts in understanding and studying society. This includes a deep dive into how technological transformations influence our lives, such as private and social interactions, education, governance, democratic processes, and business. As the scale and dynamics of these technological changes evolve, so does the methodology employed in social sciences, especially in qualitative research. Research in the 1990s began to delve into the implications of digitality and digital interactivity. Scholars explored the immediacy and omnipresence of digital communication, the interactive and participatory characteristics of digital media, and the trend towards "shallow" information searches that are quick and surface-level. They may not dive deep into the topics. These discussions share roots with Postmodernism, acknowledging the media's profound influence on identity, culture, and societal structures. They follow the tradition of postmodernism, assuming that media plays a crucial role in forming personality, culture, and social order; they diverge significantly from analogue critical theory. They highlight a departure from traditional analogue critical theories in that audiences can produce new texts that support the actions of other participants rather than just their idiolect, implying that in the digital age, everyone has the potential to be a creator or influencer. In this digital age, audiences are not just passive interpreters of the content; they actively create new content, influencing and shaping the behaviours and perspectives of others.

Today, digitality primarily manifests in the ability to store, search, and categorise information, exemplified by tools and platforms such as the World Wide Web, Google search engines, and Big Data repositories. It also facilitates communication via mobile

phones, blogs, vlogs, YouTube, and email. However, the digital era has not come without its drawbacks. Issues like computer viruses, loss of anonymity, the spread of fake news, and spam emails plague this information age. Thanks to digital possibilities and advances, our society, economics, and culture are changing and revolving in ways we have never seen before. Mobile technologies, Cloud collaboration, Big Data analytical systems, Natural Language Processing, Text Mining, Neural Networks algorithms, and the Internet of Things offer incredible and unseen individual and social opportunities, driving economic growth, improving citizens' lives and efficiency in many areas of our lifeworld, i.e., transforming the way we live and interact. These innovations drive economic progress, enhance the quality of life for individuals, and boost efficiency across diverse sectors, from education and health services to transportation, energy, agriculture, manufacturing, retail, and public administration (Schallmo and Tidd 2021). Beyond these tangible benefits, digitality also plays a transformative role in governance and policymaking. Digital tools empower policymakers with data-driven insights, fostering more informed decision-making. Furthermore, these technologies stimulate citizen engagement, promote greater transparency, and enhance the accountability of governing entities. Notably, the widespread accessibility of the Internet holds the promise of strengthening democracy, championing cultural diversity, and safeguarding fundamental human rights, such as the freedom of expression and access to information and connectivity.

To better understand how these technological changes affect our social and private life, education, science, government, democracy, or business, we must also understand how their scale and dynamics affect the contemporary research methodology of social sciences, including qualitative methods. This highlights the interplay between technology and research methodologies. The digital revolution demands that we critically evaluate changes in data collection techniques, analysis procedures, and qualitative theorizing. Discerning the potential advantages and challenges of digitizing qualitative research and computer-assisted analytical practices is crucial. Indeed, the rise of digital methodologies has opened the door to many qualitative digital possibilities. A case in point is the recent COVID pandemic, which, as noted by Wa-Mbaleka and Costa (2020), compelled qualitative researchers to pivot from traditional in-person methods like focus groups and interviews to online platforms. Researchers can communicate with more people in less time in more difficult-to-reach regions. Conducting interviews in the comfort of participants' homes fosters deeper intimacy. They can leverage technology to record interviews, create transcripts, and use mobile devices to organise the sequence of answers. This transition represents a shift in research methods due to technological advancements. However, as Palys and Atchison (2012) emphasise, the rapid change to online qualitative research methodologies has been met with mixed responses from the research community. Some voice concerns, citing epistemological and methodological reservations against swift digitisation and Computer-Assisted Qualitative Data Analysis Software.

Conversely, others view this as an opportunity to innovate in qualitative research and enhance their methodological and analytical prowess. This highlights the debate and differing opinions among researchers about the role and impact of technology in research. The evolution of research methodologies has always been influenced by technological advancements and the changing ways we interact with the world. The widespread adoption of digital technologies made their impact evident in academic research and exploration. This integration of technology and research opened up novel avenues, challenging traditional paradigms and forging new directions. As digital technologies became more pervasive, their influence seeped into academic research and investigation.

Around 2007, internet-related research underwent a significant transformation, often called the "computational turn" or data study (Berry 2011). This "computational turn" describes adopting new techniques and methodologies from computer science and its associated fields. At this point, the Internet ceased to be viewed as a distinct "cyberspace" or an extension of offline society. While the digital divide among Internet users persists, there is no return to "virtual" research in the old style. Instead, the Internet began to be

studied as collections of different social and cultural data, a space for communication and interactivity, which can revolutionise our understanding of collective human behaviour (Rogers 2009). In this regard, two critical articles were published: "A twenty-first-century Science" (Watts 2007) and "Computational social science" (Lazer et al. 2009). Their authors discussed how we could study societal conditions and cultural preferences with Internet data. Most research on human interaction has been based on selected data relating to thematically focused case studies. Digital technologies offer an unusual, second-by-second picture of interaction over extended periods, providing information about relationships' structure, dynamics, and content (i.e., ego or social network analysis). The research shift is from individuals to societies, individual to collective thinking, and single behaviour to social patterns, without limiting the number of study participants. The term computational turn refers to the process by which new techniques and methodologies are drawn from computer science (including interactive information visualisation, scientific visualisation, data pre/processing, geospatial representation, statistical data analysis, network analysis, natural language processing, and data/text mining), and related fields (cognitive science, machine learning, data science, and computational linguistics) are implemented in the humanities and social sciences. They help to aggregate, manipulate, and manage structured and unstructured data. In social science, two terminologies relate to this turn: Social Science Computing (SSC) and Computational Social Science (CSS). CSS is referred to as the field of social science that uses computational approaches in studying social phenomena. SSC is the field in which computational methodologies are created to assist in explanations of social phenomena.

Digital methods help us to study social change and cultural conditions using various online data leveraging technologies to gain deeper insights into societal trends and patterns. These methods can use, for example, computational algorithms embedded in digital devices or computer language objects such as HTML or XML hyperlinks, tags, timestamps, likes, shares, and retweets to learn how people communicate, their opinions, and how they behave online. Digital methods are part of a computational turn in the contemporary humanities and social sciences. They are positioned alongside other recent approaches, such as cultural analytics, cultural studies, webometrics, and altmetrics, where distinctions are made between the data types (natively digital or digitised) and the algorithms (written or implemented). Their versatility allows the development of new strategies in computer-aided qualitative data analysis. With the increased computing power of computers over the last few years, together with the growing amount of cultural data now available in digital form, computer software can analyse an unlimited number of textual and visual data contained in corpora. This is a new dimension of thinking in narrowly focused, contextual data analysis and qualitative research methodology. The field of digital qualitative research is becoming a rapidly growing multidimensional and multifaceted research area. More and more researchers are addressing the social, cultural, political, anthropological, and other dimensions of Computer-Mediated Communication (CMC) or using CMC to generate, collect, and analyse field data. Digital methods provide various research strategies for dealing with online data's temporal and unstable nature. In addition, these methods have been successfully used to identify the problems with online data, such as the unsustainability of web services and the instability of data streams, where APIs are reconfigured or cease to function. Qualitative research practices in the digital world can be supported by digital tools at every step of the research project, from the data collection, transformation, and data analysis to the outcomes. These observations combine three elements: (a) digital technologies and their possibilities ensure doing new things that the qualitative research community has never undertaken before and doing better things than it has always performed; b) the social dynamics triggered, supported, and fuelled by the development of digital technologies and the implications they have for sampling and social research; and (c) the possible implications of these social and technological changes for the development of the field of qualitative research (Palys and Atchison 2012).

### 4. Datafication, Digital Humanities, and New Analytical Approaches

The digital era brings datafication (Flensburg and Lomborg 2023), permeating the very fabric of qualitative research. The widespread digital integration between traditional and new approaches has reshaped how we understand and interpret information. This constant influx of data derived from numerous digital sources challenges traditional methodologies and beckons for novel approaches to qualitative inquiry. Datafication—the constantly increasing amount of data in daily life and the improvement of analytical techniques (new algorithms, new functionalities, more efficient software)—represents a paradigm shift in our analytical mindset (Dijck 2014). This makes theorising redundant in discovering knowledge about the regularities that govern society. In essence, we are transitioning from assumption-based models to data-driven understandings. Big Data, Digital Humanities, and other new approaches to data analysis help produce meaningful knowledge about complex social phenomena without the need to formulate hypotheses. The data are supposed to speak for themselves, free from theoretical limitations or researchers' assumptions. It is a move towards a more organic interpretation of data.

Big Data and new digital technologies help researchers focus on finding causes by looking at related data. The answer to why becomes less important than the search for the answer to what. This represents a profound shift in how we approach research questions. The aim is not to discover the causes of phenomena and processes but to look for the connections and relationships between the data, codes, categories, or concepts, as in qualitative data analysis. Without digitisation and datafication, there would be no Big Data and the modern CAQDAS, combining interdisciplinarity and multiparadigmacity in qualitative data analysis and research. The digital transformation has, in many ways, revolutionised the landscape of qualitative research. The breadth of contemporary approaches to qualitative data analysis can seem daunting even to experienced social scientists or researchers, especially as they move from the general analytic strategies used in qualitative research to more specific approaches for different types of qualitative data, including interviews, text, sounds, images, videos, and so-called virtual data. These diverse datasets call for nuanced methods of interpretation and understanding. However, general observations regarding implementing new digital technologies into the methodology of computer-assisted qualitative data analysis and qualitative research practices require comprehensive solutions regarding data archiving, data security, or computational capabilities. As we innovate, we must ensure the integrity and security of our data and methodologies.

Big Data, Digital Humanities, and new technologies introduce a novel epistemological perspective on designing and implementing social research. The rise of Big Data means that researchers are now confronted with far larger qualitative data sets than before. This shift has changed how we perceive and approach the gathering and analysis of data. Knowledge in such research is not solely derived from testing theories based on relevant empirical data. Instead, data, digital methods, and advanced algorithms, especially from Natural Language Processing (NLP), have become paramount sources of cognition. Utilizing computer data analysis, mainly via NLP tools, ensures that qualitative research is conducted more systematically and comprehensively. NLP, a subset of artificial intelligence, enhances the analysis by facilitating automated sentiment and topic evaluations. Our understanding of the social world emerges from these tools and data sources. Traditional methods of theory testing are now complemented by insights gleaned directly from the vast amounts of data processed using digital tools. Rob Kitchin astutely noted this transformation. He emphasised that this shift affects the broader scientific community, not just qualitative research. Kitchin underscored the profound implications of these evolving methodologies on the world of science (Kitchin 2014a, 2014b). We are witnessing a multidisciplinary, digital paradigm that cannot be solely defined in terms of traditional scientific cognition. This paradigm champions CAQDAS for adept data management and coding.

With the help of NLP tools, this method can easily manage data from many different sources, making more data available for qualitative research. This evolution suggests that the traditional notion of a "one-size-fits-all" approach to science is under revision.

Instead, we now have a methodology that can easily handle vast qualitative (textual and visual) data sets, bridging the qualitative–quantitative divide and fostering the rise of mixed-methods research. The digitisation process in research and analysis showcases the diverse strategies inherent in qualitative research and highlights ethical considerations concerning data privacy, consent, and transparency. As digitisation becomes integral to research, it reflects qualitative researchers' myriad techniques and perspectives, ensuring the data is responsibly collected, stored, and analysed.

In a Digital Society, qualitative research methodology can be implemented by extracting data from pre-existing digital platforms like forums, social media, and websites through web scraping techniques, or by employing digital tools designed for researchers that facilitate direct interaction with participants in their online environments. Examples of such tools include web-based software for conducting interviews and software for online interview transcription. Moreover, considering modern qualitative research, we must consider the digital possibilities opening up as these platforms and tools redefine how we interact and convey information because of the virtual revolution of portable computing power brought about by the different mobile devices like smartphones, tablets, and wearables, and also by the digital possibilities generated by business trendsetters such as Apple, Facebook, or Google, and their respective apps which have millions to billions of users, making them significant data sources. The digitally managed research process requires an understanding of the impact of digital technologies on all aspects and phases of design, implementation, coding, and analysis and the dissemination of qualitative research results, which means considering both the advantages and challenges posed by these technologies. Along with the datafication and digitisation of qualitative research and greater collaboration, their methodology is changing, adapting to the dynamic nature of the digital age and the language of data analysis and research practices (digital/online methods, virtual ethnography, hypermedia methods, and so forth), which requires clarification and classification to ensure consistent understanding and application among researchers. Using digital tools in qualitative social science research is not necessarily new but appears to be steadily increasing, highlighting the growing trust and reliance on these tools. Digitality helps to research without time and space limits, offering researchers unprecedented flexibility and reach, blurring between quality and quantity—on the way to digital mixed methods. These changes, in turn, lead us to a discussion on the validating standards of practising digital qualitative research, emphasizing the need for rigour and integrity in the digital era (Brown 2002; Dicks 2012; McCrohon 2013). Digital qualitative methodologies not only introduce challenges in data management but also raise essential queries regarding the genuineness and reliability of data obtained through digital means. Achieving a delicate balance between capitalising on digital advantages and upholding research integrity necessitates a profound comprehension and reflective implementation of these digital techniques. In facing these challenges, researchers must continuously strive to reconcile cutting-edge digital methodologies with fundamental research ethics, research principles and qualitative data security.

Ensuring robust digital data security has become pivotal in the era of datafication and digitisation, particularly in CAQDAS and qualitative research. The prevalent use of interactive collection methods, such as online surveys and computer-assisted interviews, necessitates stringent protocols to safeguard data and uphold the anonymity of respondents, especially amidst escalating concerns over cyber-attacks and data breaches. It's worth noting that technological means to secure data, which have become paramount, include robust cybersecurity measures and well-established procedures for data management and researcher training in ethical data handling. Furthermore, adherence to various data protection regulations, notably the General Data Protection Regulation (GDPR) in the European Union, is imperative, underscoring the necessity of obtaining explicit and informed consent, practising data minimisation, and having a legitimate basis for data collection, storage, and usage. Non-compliance with these regulations can result in significant fines and damage to reputation. Therefore, a clear understanding of these regulations is not just a legal ne-

cessity but also informs ethical research practices, ensuring that participant data is treated with the utmost respect and integrity throughout the research process. Anonymisation, which involves meticulously altering personal data to prevent the identification of subjects without additional information, emerges as a crucial tool. Researchers, therefore, must ensure that data, even when stripped of identifiable markers or replaced with pseudonyms, remains thoroughly anonymous and is immune to reverse-engineering tactics that could compromise participant identity. Employing technological solutions to bolster data security, such as frequent software updates, the use of strong, unique passwords, and the application of encryption in data transit and storage, is not merely an operational requirement but also an ethical obligation. Moreover, the implementation of these technological and procedural safeguards must be transparently communicated to participants, ensuring that they are fully aware of how their data will be protected throughout the research and beyond. Some commercial CAQDAS programs, including Atlas TI, NVivo, Maxqda, and webQDA, offer solutions facilitating collaborative analysis, like client-server or cloud-based working spaces, enabling researchers to collaborate without jeopardising data security. However, ensuring data security in these collaborative platforms rests with the researchers or the analytical tool providers. Thorough vetting of third-party providers and software used in the research process is essential to mitigate risks. Prudent selection of digital platforms for research, hence, not only safeguards data but also enhances the reliability and validity of the research findings, ensuring that they are derived from a secure and stable digital environment. While desktop solutions remain available, the burgeoning phenomenon of datafication and associated computational demands deem traditional data storage and analysis methods progressively unreliable. Consequently, researchers seek more potent and secure data-handling solutions. In conclusion, as the digital landscape continues to evolve, researchers must balance leveraging advanced digital and collaborative tools and maintaining rigorous data protection, consistently placing the ethical treatment of participant data at the forefront of their practices.

## 5. The Importance of Researcher's Digital Skills

We can access various innovative tools and methodologies in today's digital age. However, their practical use is less about the tools themselves and more about the professional skill set that wields them. These skills, encompassing technical, soft, and ethical dimensions, form the bedrock of quality research in our digital era (Janssen et al. 2013; Oberländer et al. 2020; Pope and Costa 2023).

A primary component of this skill set is technical proficiency. Modern professionals should be familiar with various software and platforms and adept at leveraging their intricate features and functionalities. Alongside this, technological know-how is a crucial aspect of digital literacy. This goes beyond just using digital tools; it involves discerning which tool is best suited for a specific task and evaluating the credibility of online resources. While AI tools bring advanced capabilities, they are not without flaws (Costa 2023). This is where the skill of AI interpretation becomes indispensable. Professionals must be competent in reviewing AI-generated outputs, identifying potential inaccuracies, and ensuring precise products, like transcriptions. Understanding cybersecurity fundamentals is paramount, given the increasing cyber threats in our digital-centric world. Professionals should be versed in best practices related to data encryption, secure data storage, and safe data transmission.

Soft skills, which can sometimes be undervalued next to technical skills, are equally essential. In the realm of virtual interactions, active listening becomes vital. Professionals must be attuned to subtle vocal nuances, pauses, and inflexions without physical cues. Effective communication is also paramount, especially in virtual settings where physical cues are lacking. Moreover, the rapid advancements in the digital world necessitate professionals to be adaptable, ready to learn, and pivot as new tools and methodologies come to the fore. As digital tools erase geographical boundaries, cultural awareness and empathy become more critical. Professionals must navigate these global interactions sensitively,

understanding the varied cultural nuances to foster genuine exchanges. Efficient project management is another crucial skill, especially when dealing with digital resources, virtual teams, and online tasks. This ensures that projects progress seamlessly, even in dispersed digital environments. Ethics is pivotal in the digital age, accompanied by unique data privacy and transparency challenges. Professionals must maintain integrity and uphold stringent ethical standards. Coupled with this is the commitment to ongoing learning. Professionals should stay updated via regular workshops, webinars, and training sessions as the digital landscape evolves. Of course, collaboration, too, is essential. While digital tools have made it easier to bridge geographical divides, they also require a solid collaborative spirit to ensure smooth teamwork, even when teams are globally dispersed. Finally, the significance of feedback in this digital age cannot be overstated. Professionals should proactively seek, analyse, and utilise such input as a driving force for methodological refinement and overall growth.

In an increasingly digitised world, where almost every facet of daily life intersects with technology, mastering the tools of CAQDAS and digital methods becomes imperative for qualitative researchers. However, a distinct disparity in the grasp of analytical and IT skills is evident among many in the field (Mertens et al. 2017; Torrato et al. 2023). This mismatch impedes fully realising digitisation's potential in the social sciences. This potential, which lies in processing vast amounts of data and identifying patterns at a pace unimaginable a few decades ago, adds a layer of depth to research. The initial expectation for researchers might have been a primary focus on their study area, but the rapid pace of technological advancements has set a new paradigm. This shift has moved from pen-and-paper data analysis to a reliance on complex digital tools and software. Many researchers without formal training in IT or computer sciences find themselves on a steep learning curve. Navigating this curve requires patience, determination, and often a willingness to venture outside one's comfort zone. The self-learning process (Freitas et al. 2018a, 2018b) might be daunting, but it is propelled forward by these researchers' innate curiosity and analytical prowess. By immersing themselves in the dynamic realm of technology, researchers bridge the competency gap and bring innovative solutions that defy the constraints traditionally associated with digitisation. These innovative solutions might range from new data visualisation techniques to implementing machine learning algorithms in qualitative research. This technological era has emphasised the pivotal role of interdisciplinary collaboration, particularly between IT and qualitative research. Melding the methodologies and tools from fields like Digital Humanities, Corpus Linguistics, Big Data Analysis, and Computer Science has redefined the contours of qualitative data analysis. This fusion allows for harnessing machines' computational power and precision with a nuanced understanding of human behaviour, facilitating deeper research explorations. This amalgamation enriches the research process, allowing for more profound insights and broader applications. Such expanded scope is akin to opening a previously locked door in the mansion of knowledge. Yet, even as the horizons expand, challenges persist.

For instance, the realm of collaborative technologies or the intricacies of database systems often remains enigmatic for many qualitative researchers. In the digital age, a researcher is not merely expected to possess expertise in data analysis methodologies or field experience; their skillset now must encompass database management and even touch upon programming. They are expected to wear multiple hats, transitioning seamlessly from a core researcher to a pseudo-technologist. Integrating CAQDAS programs with languages like Python or R is a testament to this shift. Such advancements cater to tasks like data pre-processing, automatic coding, and deeper analyses. However, lacking these nuanced skills compels qualitative researchers towards more collaborative avenues. Recognizing one's limitations and seeking partnerships becomes crucial. Teaming up with professionals from diverse domains like computer science and mathematics fills the knowledge gap and brings a confluence of perspectives to the research, enriching it manifold. In essence, the future of qualitative research hinges upon a harmonious blend of traditional methodologies

and the evolving digital toolkit. Embracing this change, researchers stand at the cusp of a revolution in how qualitative data is collected, analysed, and interpreted.

## 6. CAQDAS, Digitality, and Collaborative Working

We live in digital culture, meaning digitality encourages us to connect, collaborate, communicate, and participate in global social networks. One of our "digital culture" core beliefs is that digital networks encourage excellent connectivity, collaboration, communication, community, and participation. This can be seen in social and news media discourse (Facebook, Twitter, YouTube, blogs, peer-to-peer TV/Internet networks, Netflix platforms, open-source software, etc.). But, it becomes even more visible and tangible in the computer-aided analysis of qualitative data software, which is becoming more collaborative, systematic, and interactive. In social sciences, there is a shift from the digitality of qualitative research to collaborative data coding, analysis, and thinking (Uehara et al. 1996; Richards and Hemphill 2018). We have commercial and online software for collaborative data collection, collaborative data coding, collaborative analysis, collaborative thinking, collaborative writing, etc. The qualitative research process can be conducted digitally and collaboratively on the web (server or cloud computing) or desktop software: interview transcribing, project and task managing, codebook preparing and coding, data analysis and modelling, interpretation and theorising, and final writing and representing findings. With the digitalisation of qualitative research and the prevalence of CAQDAS among researchers, a new style of thinking and approach in qualitative data analysis is taking shape based on collaborative methodology and teamwork.

There has been a growing interest in research collaboration in recent years, and different terminologies have emerged to describe this phenomenon. In a descriptive literature review conducted by Yang and Tate (2012), they explored the field of cloud computing research and proposed a classification structure. Similarly, other scholars have been inspired by the potential of web-based collaboration and have used terms similar to those employed in this study. For example, Bröer et al. (2016) introduced the concept of collaborative interpretation, which involves researchers working together to interpret and analyse data, leveraging the power of online collaboration tools. This approach acknowledges the benefits of collective intelligence and the diverse perspectives that can be brought to the interpretation process. Another related term is online collaborative research, which refers to research conducted openly and collaboratively, often leveraging online platforms and communication tools. This approach embraces the principles of openness, inclusivity, and shared knowledge creation, allowing for greater engagement and participation from a diverse range of researchers. These terminologies reflect the evolving nature of research collaboration and the increasing reliance on digital technologies to facilitate collaboration and knowledge sharing. The emergence of cloud computing and web-based platforms has expanded the possibilities for collaboration beyond geographical boundaries, enabling researchers to connect and collaborate globally. It is important to note that while the specific terminologies may vary, the underlying principles and objectives remain consistent. The goal is to enhance research collaboration, foster innovation, and leverage the collective expertise of researchers to advance knowledge and address complex challenges. These terminologies offer valuable insights into the various dimensions of research collaboration and provide a foundation for further exploration and understanding in this rapidly evolving field.

The literature review shows that collaborative analysis and research can be carried out on three primary methodological levels: interdisciplinary (Makel et al. 2019; Frost et al. 2010; Tartas and Muller Mirza 2007), international (Akkerman et al. 2006; Marková and Plichtová 2007), or interpersonal: senior–junior (Hall et al. 2005; Rogers-Dillon 2005), insider–outsider (Louis and Bartunek 1992), and academic–practitioner (Hartley and Bennington 2000). Collaborative analysis of qualitative data seems to hold a variety of effects, from a more informed, complex, or helpful digitally supported qualitative data analysis leading to new interpretations, transcending present knowledge, or creating possibilities for

individual learning or improving new analytical skills (Cornish et al. 2014). Such potential benefits are not risk- or cost-free. Risks and costs, like the benefits, are derived from the confrontation of diverse perspectives and research methodologies. In that case, collaboration needs institutional support and flexibility, straightforward working procedures, and social relations, promoting open research debate without threatening the researcher's identity. All may help to alleviate the potential risks of collaborative analysis and research. Of course, effective collaboration in qualitative research depends on the classes of research and analytical methods applied, the number of people participating in the project, or the project scale. The digitisation that permeates our daily lives further blurs the distinctions between commercial software programs. These now boast enhanced analytical capabilities and foster a more collaborative environment for research (Costa 2016). The digital transformation has reshaped the dynamics of scientific work, drawing it closer in nature to conventional business projects.

Emphasizing the pivotal role of digital tools, this article delves into the evolution of computer-assisted qualitative data analysis. It underscores the importance of collaboration and harnesses digital advancements in modern qualitative data collection and analysis. The growing evidence suggests that digital tools are invaluable in bolstering collaborative efforts and refining the qualitative research process. For instance, Costa et al. (2016) introduced the 4C collaborative work model, outlining the collaborative capabilities of the qualitative analysis software webQDA. Echoing the importance of collaboration, Davidson et al. (2016) stressed the necessity for thriving communities of practice. These communities are crucial in facilitating the development and application of digital tools in qualitative research. Furthermore, Crichton (2012) posited that digital tools not only streamline the tasks of qualitative researchers but also enrich the data, offering more significant depth. Reinforcing this viewpoint, Paulus et al. (2014) delivered a comprehensive overview, showcasing how digital tools can be leveraged at various junctures of the research journey.

In practice, research collaboration and collaborative analysis have numerous methodological advantages. Three notable benefits are analytical credibility, methodological reflexivity, and intersubjective thinking. Credibility is a synthetic outcome of the primary analytical process stages, including data coding, investigating the code's relationships, developing interpretations, and qualitative theorizing. Measuring methodological reflexivity or intersubjective thinking presents a challenge and becomes more apparent in later data analysis stages than credibility. Thus, we can demonstrate what does not work within the team coding process. Coding is at the core of qualitative analysis, but its effectiveness depends on the size of the volume of data we have. This is an iterative process in which the structure of the codes is dynamic and undergoes a continuous transformation with the researchers delving into semantic contexts and the semantic structure of data. With the digitisation of qualitative research and the development of new CAQDAS functionalities, greater emphasis is placed on the data validation procedure and ensuring the reliability of coding (Lu and Shulman 2008; Sweeney et al. 2013; O'Connor and Joffe 2020). To verify this reliability, we use the inter-coder agreement procedure. Computing the compatibility of coding is used to compare coding consistency between several coders. Such a procedure can help uncover the differences in interpretation, clarify equivocal rules, identify ambiguity in the text, and finally quantify the level of agreement obtained by coders. In practice, we can uncover the differences in interpretation, clarify ambiguous rules, identify ambiguity in the text, and ultimately quantify these coders' final level of agreement.

Unfortunately, applying inter-coder agreement procedures often involves requirements or assumptions incompatible with the qualitative data analysis processes. At least two compatibility problems can be identified: the codebook problem (using codes) and the segmentation problem (applying codes to texts). Generally, the CAQDAS software may use the four inter-coder agreement criteria based on the code occurrence, frequency of using code, importance (text covered by code), and overlapping codes in a text (two codes cover the same piece of text). The methodological, collaborative reflexivity is overcoming individual rationality to collective rationality and incorporating local understanding

(common thinking) into global understanding (scientific thinking). This involves the third aspect of collaboration, creating an intersubjective space for open dialogue, discussion, and perspective-transcending knowledge. This process may be described as collaborative knowledge production in computer-assisted qualitative data analysis and digitally supported research. In a collaborative context, the researcher must be willing to work in a framework of mutual support between peers and participate in the synergy of the group to organise complex tasks via communication. The collaborative process offers, in particular, the possibility of interacting effectively and allows the development of analysis, synthesis, problem-solving, and evaluation skills. Thus, as a source of encouragement and support, the collaborative process presents itself as a means of learning and enrichment, in which the sphere of collaboration does not supplant the sphere of action of the individual. However, for the researcher to adopt this attitude, they must see themselves in the collaborative approach and have the means to enable, promote, and facilitate collaboration.

## 7. Summary: Going Digital and Collaborative but Staying Qualitative

As the wave of digitisation and datafication intensifies, the domain of social sciences, particularly the qualitative research methodology, is evolving into a predominantly data-driven sphere. Intriguingly, this shift aligns with the core principles of grounded theory. What is paradoxical about this transition is that data scientists now frequently explore questions that were once the exclusive domain of sociologists. These data specialists utilise vast datasets and employ methodologies that diverge considerably from the conventions of social sciences. Such a shift exposes traditional social sciences, such as sociology and anthropology, to the potential risks of becoming overshadowed. This looming risk is amplified by the surge of digital research techniques demanding advanced computational knowledge. Further compounding the situation is the heightened rivalry from the corporate realm, which often has superior access to data. This potential sidelining is especially pertinent when discussing qualitative research. However, it is crucial to underscore that sociologists and anthropologists, unlike many data scientists, possess a rich history and expertise in qualitative research. With the exponential growth of quantitative datasets, extracting meaningful insights without integrating qualitative methodologies becomes a formidable challenge. Thus, the current landscape underscores the importance of "Thick Data" amidst the prevailing "Big Data" epoch (Jemielniak 2020).

The ascendency of digitality and digital technology is fundamentally altering the practice of qualitative research and computer-assisted data analysis. Becoming proficient in qualitative research methodologies and computer-assisted data analysis is closely linked to grasping and applying new digital technologies and enhancing digital skills. The mediation of research and analysis via digital technology is becoming a norm, subtly shifting perceptions of qualitative analysis and its execution. These changes resonate with our research initiatives' epistemological and ontological foundations. In the past decade, qualitative research, especially multimedia digital data, has benefited from developing and advancing software tools that support most core qualitative methodological techniques. Contemporary qualitative research is no longer limited to small interviews but calls for concerted cooperation among researchers. The digitalisation of qualitative research and the growing prevalence of CAQDAS is forging a new paradigm in qualitative data analysis built on collaboration and teamwork (Seror 2012). CAQDAS and other new digital tools illustrate how technology can bolster the research process, offering time efficiency and adding substantial depth to qualitative work. They facilitate every phase of the research process, drawing on various tools, possibly already familiar to many researchers and providing practical case studies drawn from actual research. Whether we use traditional or digital, computer-assisted methods, it is essential to recognise that qualitative data analysis mandates the careful, systematic, and thorough management of substantial text data, such as interviews, notes, and internet data. Thus, the prerequisite for reliable qualitative analysis is efficient and consistent data management, for which adopting digital technologies and appropriate CAQDAS software is natural and obvious.

Qualitative analysis, which is inherently complex and multifaceted, begins with fieldwork and involves a carefully planned sequence of activities: conducting interviews, transcribing recordings, reading transcriptions, retrieving phrases, coding text and images, analysing data and visualising it. Due to the nature of this fieldwork flow, CAQDAS and other digital solutions facilitate a seamless transition through the various stages—from the subtleties of the transcription process to the complexities of data analysis to the formulation of theory. Software and digital tools such as CAQDAS empower researchers by streamlining the transcription process, enabling collaborative research, and supporting the development of robust qualitative analysis models. In the broader perspective, integrating digital tools into qualitative research is not a mere convenience but a necessity. The objective is clear: to attain a nuanced comprehension of the data and rigorously evaluate the effectiveness of the analytical strategies employed. As we delve deeper into the digital age, digitisation, collaboration, and relationality unmistakably define the essence of contemporary computer-aided qualitative data analysis and qualitative research methodology on a broader scale. The transformative effects of digitisation, datafication, and innovative technologies have been profound. They have reshaped qualitative research's data collection, processing, and analysis methodology. With the advent of these modern information and communication technologies, a fresh era is unfolding—one marked by unprecedented interconnectedness, presenting researchers with diverse and previously unthought-of opportunities. This new era, empowered by these transformative technologies, has transcended the traditional confines of time and space, forging pathways for enhanced collaborative endeavours in research. Furthermore, it is worth noting the democratizing influence of these digital tools on the research landscape. Their ability to tap into vast virtual networks allows qualitative researchers to observe and actively immerse themselves in the investigative process.

The spirit of collaboration, rooted in mutual reliance, collective synergy, and shared objectives, drives more successful and impactful research outcomes. Yet, an evident paradox exists: while the technological potential is boundless, its full exploitation is hampered by researchers' reticence. Whether overlooking the available software or an inability to fathom the collaborative potential of these tools, the root cause often traces back to a fundamental unfamiliarity and lack of expertise with these digital resources. This disconnect between the technological possibilities and their adoption is glaring. To bridge this chasm, a dual-pronged strategy is essential. Practically speaking, amplifying awareness about these tools' multifaceted capabilities and dependability is urgently needed. On a cultural front, the research community must evolve, transitioning from the siloed, individual-centric research ethos to a more inclusive, dialogic model that is holistic in its approach.

In conclusion, the emergence of novel information and communication technologies has enhanced networking capacities and revolutionised collaborative work in qualitative research and computer data analysis. Nevertheless, challenges arise due to researchers' insufficient knowledge and utilisation of these tools. Via focused instrumental and cultural interventions, researchers can fully harness the transformative potential of these technologies, aligning with the prevailing trends of datafication, digitisation, and collaboration that underpin computer-assisted qualitative data analysis and qualitative research methodology.

**Funding:** The work of the first author was entitled "*The domain ontology as a model of knowledge representation about the contemporary field of qualitative research*" and financed from 2017 to 2021 by Narodowe Centrum Nauki (Poland, Project No. 2016/23/B/HS6/00301). The work of the second author is funded by national funds through FCT—Fundação para a Ciência e a Tecnologia (Portugal), under the Scientific Employment Stimulus—Institutional Call—[CDL-CTTRI-248-SGRH/2022] and the CIDTFF (projects UIDB/00194/2020 and UIDP/00194/2020).

**Institutional Review Board Statement:** Not applicable.

**Informed Consent Statement:** Not applicable.

**Data Availability Statement:** Not applicable.

**Conflicts of Interest:** The authors declare no conflict of interest.

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
