# Peer review of "Qualitative Research in Digital Era: Innovations, Methodologies and Collaborations"

_socsci, doi:10.3390/socsci12100570_

Round 1

Reviewer 1 Report

The article attempts to explore the impact of digital technologies on qualitative research but needs to provide a comprehensive and coherent analysis. The author presents a scattered and disjointed discussion that needs more clarity and depth. The arguments could be more specific and provide concrete evidence or examples to support the claims.

The article needs a more precise structure and effectively engages with the existing literature. The references to CAQDAS and digital tools need to offer meaningful insights into their practical application or potential limitations. Suggest adding and discussing: https://doi.org/10.3390/educsci11070361; https://doi.org/10.3390/mti6090083

The language used is overly general and lacks precision. The article must address the challenges and complexities of integrating digital technologies into qualitative research. It overlooks the ethical considerations and potential biases associated with digital data analysis.

The article needs to be more rigorous and make a compelling case for the transformative power of digital technologies in qualitative research. It requires revisions to provide a more coherent and substantive topic analysis.

na

Author Response

Dear Reviewer,

Thank you for taking the time and effort to review this text. Your feedback and comments are of great importance to us and provide invaluable input for the further development of this paper.

  1. According to the MDP classification, our text is a scientific essay and not a traditional scientific article. Thus, its purpose and nature of expression differ. An essay focuses on reflection, interpretation, and the author's subjective statement, whereas an article is more informational, fact-based, and has a more objective character. Nevertheless, we are deeply grateful for all the constructive comments and suggestions that will allow us to improve the essay.
  2. We especially appreciate the pointers regarding structure, coherence, and readability. Your comments provide a fresh and enriching perspective on our essay. With them, we better understand the direction we should take.
  3. In the essay, we rely on methodological knowledge as well as our own, over twenty years of analytical experience in the area of CAQDAS, familiarity with, and use of computer software in qualitative data analysis. Thank you for drawing attention to the ethical issues, which indeed are absent in the essay. It wasn't our original intent, but your comment on incorporating the poetic thread in the essay is well-taken.
  4. The essay primarily represents our position in the discourse on the influence of IT on the way modern computer-assisted qualitative data analysis is conducted. We completely agree that for greater clarity, our arguments should be more specific or supported by concrete evidence or examples.
  5. We hope that the improved structure of the text and the argumentation will ensure better clarity and coherence in the essay. Our aim is for the revised essay to clearly and effectively present our stance in the discourse on the impact of new technologies and digitization on the process of qualitative data analysis.

Authors

Reviewer 2 Report

The authors define and describe advances in the field of Computer-Assisted Qualitative Data analysis (CAQDAS) including real-time collaboration, datafication, and data security. The research topic is timely and of practical importance to many researchers. However, the manuscript itself lacks a clear organizational structure and key elements required to demonstrate the rigor of the systematic review conducted. I make several suggestions to improve these elements below.

1.      Although this is a non-empirical article, it would still improve the clarity and focus of the manuscript to state and use research questions to organize the systematic review.

2.      Information on the process by which the authors conducted the systematic review detailed is necessary to evaluate the rigor of subsequent findings.

3.      I am confused by footnote 1. Are the authors stating the study summarized in this manuscript is also under peer review elsewhere? If so, the authors should choose the best outlet and withdraw its submission from the other journal to avoid unethical behavior and self-plagiarism.

4.      The first paragraph of the introduction starts too narrow by talking about the lack of geographic constraints afforded by technology. I advise the authors to use this first paragraph to situate their topic in larger social and research trend instead to motivate the importance of the study.

Thank you for the opportunity to review this manuscript. I hope my thoughts are helpful as you continue refining your manuscript.

Author Response

Dear Reviewer,

Thank you for taking the time and effort to review this text. Your feedback and comments are of great importance to us and provide invaluable input for the further development of this essay.

  1. We greatly appreciate the suggestion regarding structuring the essay using research questions. We consider it a valuable and relevant insight that will help better organize the content in this essay. We are also contemplating the idea of structuring the essay with a combination of question-answer and thesis-argumentation formats to make it more discursive. This will enable readers to better understand the essay's purpose and focus on key issues.
  2. Regarding Footnote 1, we would like to emphasize that this essay has not been previously published or submitted for publication. Therefore, there is no possibility of self-plagiarism.
  3. We thank you for the suggestion to contextualize the topic within a broader social and research context, moving beyond the issue of geographical constraints. We understand this comment as an effort to anchor the essay's theme and our argumentation in a broader discourse field concerning the impact of new technologies on qualitative data analysis.

Once again, we sincerely thank you for the constructive feedback on this text, which will undoubtedly contribute to improving the quality and coherence of this essay.

Authors

Reviewer 3 Report

Dear Author(s), 

Thank you for your manuscript, which I read with great interest. You raise and elaborate on a very important area for consideration. Might I suggest some additions to Section 5: CAQDAS, Digitality and Collaborative Working? 

Much of what is discussed here has links with digital literacy i.e. collaboration using digital tools, dissemination between groups, information and data literacy as a competence for gathering, filtering and storing online data. It would be helpful if you incorporated, or acknowledged these aspects of digital literacy in this section. 

Author Response

Dear Reviewer,

Thank you very much for your thoughtful and insightful review of our manuscript. We truly appreciate your positive feedback and your suggestions for enhancing Section 5: CAQDAS, Digitality, and Collaborative Working. Your suggestion to include aspects of digital literacy in that section is indeed valuable and aligns well with our aim to comprehensively address the topic. We will certainly incorporate these suggestions into our essay. Your perspective on the links between the discussed concepts and digital literacy provides a fresh angle that will undoubtedly enrich the content and contribute to a more robust discussion. We are committed to making the recommended improvements and believe that your insights will significantly enhance the overall quality of our paper.

Authors
